# Study on Assessment Method of Failure Pressure for Pipelines with Colony Corrosion Defects Based on Failure Location

Hao Zhang [1], Mingming Sun [2,3,4,*], Jie Zhang [5], Yiming Zhang [2,3,4], Bin Li [2,3,4] and Kejie Zhai [2,3,4]

1   College of Architecture and Civil Engineering, Xinyang Normal University, Xinyang 464000, China; zhanghao@xynu.edu.cn
2   School of Hydraulic and Transportation, Zhengzhou University, Zhengzhou 450001, China; jlu631105@163.com (Y.Z.); 13523519067@163.com (B.L.); kejiezhai@zzu.edu.cn (K.Z.)
3   National Local Joint Engineering Laboratory of Major Infrastructure Testing and Rehabilitation Technology, Zhengzhou 450001, China
4   Collaborative Innovation Center of Water Conservancy and Transportation Infrastructure Safety, Zhengzhou 450001, China
5   Department of Hydraulic Engineering, HENAN Vocational College of Water Conservancy and Environment, Zhengzhou 450008, China; zzu20080419@163.com
*   Correspondence: mingmingsunyan@163.com

**Abstract:** Evaluating the burst pressure of corrosion cluster defects necessitates considering the interaction of contiguous defects. The importance of this interaction cannot be emphasized enough, as it plays a crucial role in determining the failure pressure of such pipelines. Current methods for assessing the failure pressure of corrosion cluster defects have drawbacks, including complex evaluation procedures and limited applicability. In this research, the failure mode and location of corrosion clusters with two or more defects are studied with a burst experiment on a full-scaled pipeline. Based on the failure position of the corrosion cluster, a "center failure location" method is proposed to estimate the burst pressure of colony corrosion defects. The method takes the defect in the failure position as the center; the influence of contiguous defects that interact with the central defect on its failure pressure is considered, and subsequently, the burst pressure of colony corrosion centered on the failure location is evaluated. In contrast with the current assessment methods, while the proposed approach does not reduce prediction errors, it requires fewer evaluation conditions and is operationally simpler and more versatile.

**Keywords:** corrosion cluster defects; failure location; assessment method; failure pressure

## 1. Introduction

Oil and gas are often transported via steel pipes due to their widespread utilization [1,2]. Corrosion can significantly reduce the wall thickness and load capacity of pipelines [3]. C Wang and MF Hassanein et al. [4] compared and analyzed the corrosion simulation methods for oil and gas pipelines, and the results showed that corrosion was the result of the coupling effect of multiple factors. With an increase in corrosion range, the pipelines result in burst failure [5]. The precise prediction of failure pressure is very important for the bearing capacity evaluation of oil and gas transmission pipelines [6,7].

If the two defects are close enough, they will interact with each other and form a corrosion defect cluster [8–10]. The load-bearing capacity of a pipeline with a corrosion colony is typically lower than that of the same pipeline with individual defects [11,12].

Typical interaction rules are4 described by DNV-RP-F101 (DNV GL AS 2015) [13], API 579 (2nd edition 2007) [14], BS 7910 (2nd edition 2005) [15], Kiefner and Vieth [16], and the Pipeline Operator Forum (POF) [17]. According to the finite element (FE) results, Al-Owaisi et al. [18], Li et al. [8], and Benjamin et al. [19] all provided the modified evaluation criteria. These criteria showed superiority in judging the interaction between defects.

These rules made use of criteria defined by parameters related to the cross-section of the pipe or to the geometry of the interacting defects.

PETROBRAS has conducted burst tests on adjacent defects with the same size [20,21] and different sizes [22,23], providing valuable data for the assessment of failure pressure and the finite element verification of pipelines with interacting defects. FE models of tubular X-joints were analyzed by Nassiraei, Hossein and Zhu, and Lei et al. [24], and a theoretical method was proposed to predict the ultimate capacity of X-connections. Chouchaoui and Pick [25,26] and Sun and Cheng [27,28] analyzed a corrosion cluster with different arrangements. Andrade et al. [29], Sliva et al. [30], and Benjamin et al. [20] adopted a numerical simulation to analyze the failure mechanism of contiguous defects. Chen et al. [31] developed a failure pressure evaluation process suitable for the X80 pipeline with circumferentially or axially aligned defects. Elder Soares et al. [32] analyzed the bearing capacity of a pipeline with adjacent defects under the combined impact of thermal and pressure loads. H Nassiraei and L Zhu et al. [33] investigated the effect of the collar plate on the static capacity of circular-hollow-section (CHS) X-joints, and the results showed that the collar plate could increase the ultimate strength. The results indicated that the impact mechanism of the interaction related to failure pressure was complicated, and it depended on several factors, including the shape, size, arrangement, and spacing of the defects.

At present, there are three commonly used failure pressure assessment methods for adjacent defects: MTI [34], DNV-RP-F101 [13], and a new method put forward by Li et al. [35]. The three assessment methods equate the interacting defects with a single defect for evaluation and take the axial and circumferential ranges of the interacting defects region as the length and width of the new equivalent defect. The primary distinctions among the three methods lie in their computational models for determining the effective depth.

Although the aforementioned three assessment methods present relatively accurate predictions of failure pressure [19,34–36], their evaluation steps are more complex. Therefore, a more concise and efficient evaluation procedure is necessary. Based on the failure location of corrosion cluster defects, a "Central Failure Location" method for failure pressure evaluation is proposed in this paper. Contrasted with the current estimating codes, the calculation program of the newly proposed method was simpler and more efficient with the same prediction error.

## 2. Failure Mode Analysis of Colony Corrosion Defects of Different Types

The assessment method proposed in this paper involved estimating the burst pressure based on the failure position. Therefore, it was necessary to judge the failure positions of corrosion clusters of different types.

### 2.1. Failure Location of Colony Corrosion Defects with the Same Size

Chouchaoui conducted burst tests on an X46 pipeline with corrosion clusters [25]. The colony corrosion defects with interactions were arranged into an axial alignment. The spacing between the two neighboring defects and the number of defects is different in different cases. The specific data are shown in Table 1. It is clear that the failure modes of axially arranged colony corrosion composed of defects with same size under internal pressure are that the defects in the center of the corrosion region or middle area fail firstly, resulting in the damage of the whole pipeline.

### 2.2. Failure Location of Colony Corrosion Defects of Different Sizes

Defects with minimal internal pressure are the most probable failure positions. In order to verify the correctness of this conclusion, other tests were selected for verification. The failure characteristics of corrosion clusters with multiple interacting defects were studied.

Fifteen sets of data related to the burst tests of pipelines with multiple interacting defects conducted by the PETROBRAS R&D Center were collected to validate the failure position of corroded pipelines [22,23]. The failure model is illustrated in Table 2. The failure pressure of defect 1 was the smallest and that of defect 2 was the largest. All the tests failed

at defect 1 with the smallest failure pressure and tear to the entire corroded area under blasting pressure. The typical cases were IDTS 17 and IDTS 23, as shown in Figure 1.

**Table 1.** Geometric parameters of corrosion defects [25].

| Case | Distribution Sketch | | Failure Analysis | |
|---|---|---|---|---|
| | Arrangement Mode | Defect Number | Failure Location | Characteristic |
| S2lo-1 | | 1  2 | defect 1–2 | Failure in the middle area |
| s3lo-2 | | 3  4  5 | defect 4 | Failure in the middle defect |
| s4lo-2 | Axially aligned | 3  4  5 | defect 4 | |
| s2lc-2 | | 3  4  5 | defect 3–4–5 | Failure in the middle defect |
| s3lc-1 | | 1  2  3 | defect 1–2–3 | |
| s4lc-2 | | 5  6  7  8  9 | defect 7 | Failure in the middle defect |

**Table 2.** Number of defect 1 to defect 4 in corrosion cluster [22,23].

| Case | Defect 1 | Defect 2 | Defect 3 | Total Number | Failure Mode |
|---|---|---|---|---|---|
| IDTS 15 | 2 | 1 | - | 3 | |
| IDTS 16 | 2 | 2 | - | 4 | |
| IDTS 17 | 3 | 2 | - | 5 | |
| IDTS 18 | 4 | 1 | - | 5 | |
| IDTS 19 | 5 | 1 | - | 6 | |
| IDTS 20 | 5 | 1 | - | 6 | |
| IDTS 21 | 6 | 1 | - | 7 | Failure of defect |
| IDTS 22 | 6 | 2 | - | 8 | 1 that expanded |
| IDTS 23 | 6 | 2 | - | 8 | to the whole |
| IDTS 24 | 7 | 2 | - | 9 | corrosion region. |
| IDTS 25 | 5 | 4 | - | 9 | |
| IDTS 26 | 5 | 4 | - | 9 | |
| IDTS 28 | 1 | - | 1 | 2 | |
| IDTS 29 | 2 | - | 1 | 3 | |
| IDTS 30 | 3 | - | 2 | 5 | |

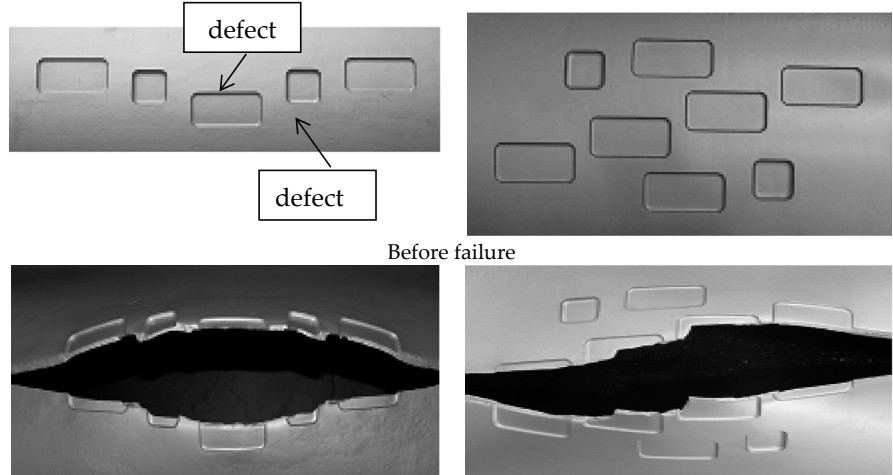

Before failure

After failure

**Figure 1.** Failure mode of the specimen [22,23].

The consistent failure modes of pipelines are observed even when there are multiple types of defects present. Failures are most likely to occur at defects with the lowest resistance. This is due to the fact that the defect is affected by stress concentration, and when the local stress increases, it results in failure. This finding was in line with burst tests previously carried out by the University of Waterloo [37] and pipelines with irregular-shaped defects [38,39].

## 3. Improved Assessment Method Based on Failure Location

The failure locations of corrosion cluster defects with different types varied according to Sections 2.1 and 2.2. To classify the failure pressure of each individual defect, a quantitative criterion needed to be defined. Li and Bai et al. [36] evaluated the failure pressure with the DNV-RP-F101 assessment code. The results showed that the average error of DNV-RP-F101 was 3.05%. Sun and Li et al. [5] evaluated 123 corroded pipelines with different strength levels and showed that the average error of DNV-RP-F101 was 10.91%. Su and Li et al. [40] evaluated 44 artificially corroded and 8 naturally corroded pipelines and showed that the average error of DNV-RP-F101 was 12%. From the comparison, we can see that for individual defect failure pressures, the deviation of the DNV-RP-F101 assessment code was around 3.05–12%, and 3% was used as a reference standard for when the burst pressures were identical.

Through the research on the failure characteristics of corrosion cluster defects, the assessment method for corrosion cluster defects based on the failure location was proposed. The evaluation procedure is presented in Figure 2, and the specific steps are as follows.

(1) In the analysis of corrosion clusters comprising defects with varying internal pressure capacities, it was observed that the likelihood of a defect becoming the point of failure increased as its internal pressure capacity decreased. Therefore, defects were ranked based on their probability of becoming the failure point from highest to lowest. In other words, the failure pressure of each defect was sorted from smallest to largest, and then the defect with the corresponding failure pressure served as the 'central failure position', in order. This ensured that every individual defect was included in the evaluation. The detailed evaluation steps are outlined as follows:

**Step 1**: Evaluate the burst pressure of isolated corrosion defects. Arrange the $N$ defects in ascending order of their internal-pressure-bearing capacity: $defect-1$, $defect-2$, ..., and $defect-N$.

**Step 2**: Select the corrosion defect $defect-1$ as the "central failure position", and identify other corrosion defects that interact with it based on predefined interaction criteria. These interacting defects, along with $defect-1$, constitute the initial evaluation object, denoted as $colony-1$. Subsequently, iteratively select $defect-2$, $defect-3$, ..., and $defect-i$ as the "central failure position" to form subsequent evaluation objects, namely $colony-2$, $colony-3$, and so forth. The final evaluation object is designated as $colony-i$, ensuring the inclusion of all single defects in the evaluation process.

**Step 3**: Proceed to evaluate the failure pressures of the cluster evaluation objects, sequentially from $colony-1$ to $colony-i$, resulting in a set of failure pressures: $P_1, P_2 \ldots P_i$.

**Step 4**: Determine the pipeline's failure pressure $P_f = \min\left\{P_{defect-1}, P_1, P_2 \ldots P_i\right\}$, where $P_{defect-1}$ is the burst pressure of $defect-1$. The predicted pressure is comprised of two components: one is the minimum value of the failure pressure for a single defect ($P_{defect-1}$), and the other is the minimum value among the corrosion cluster evaluation objects ($\min\{P_1, P_2 \ldots P_i\}$).

(2) As a corrosion cluster consists of defects with similar internal pressure capacities, defects closer to the center experience stronger interactions from adjacent defects. This increases the likelihood of it becoming the point of failure. Therefore, the defects were sorted according to the probability of being in the failure position from largest to smallest, i.e., the distance between the defect and center point from smallest to largest, and then the failure pressure was evaluated with the defect as the "central

failure position", in order. As all individual defects were involved in the evaluation, the failure pressure assessment procedure was concluded.

**Step 1**: As shown in Figure 3, determine the rectangular region of corrosion cluster defects according to the range of circumferential and axial projections of the corrosion cluster defects. Select the center point (centroid position) $O$ of the rectangular region, which can be the intersection of the axial and circumferential center lines of the corrosion region.

**Step 2**: Separately calculate the distance between the center point of every single defect and point $O$. Sort the defects according to distance from smallest to largest: $defect-1$, $defect-2$, ..., and $defect-N$. Calculate the burst pressure $P_{defect-1}$ of $defect-1$.

The remaining steps are the same as **Steps 2–4** in the evaluation procedure of the corrosion cluster defects with different internal-pressure-bearing capacities.

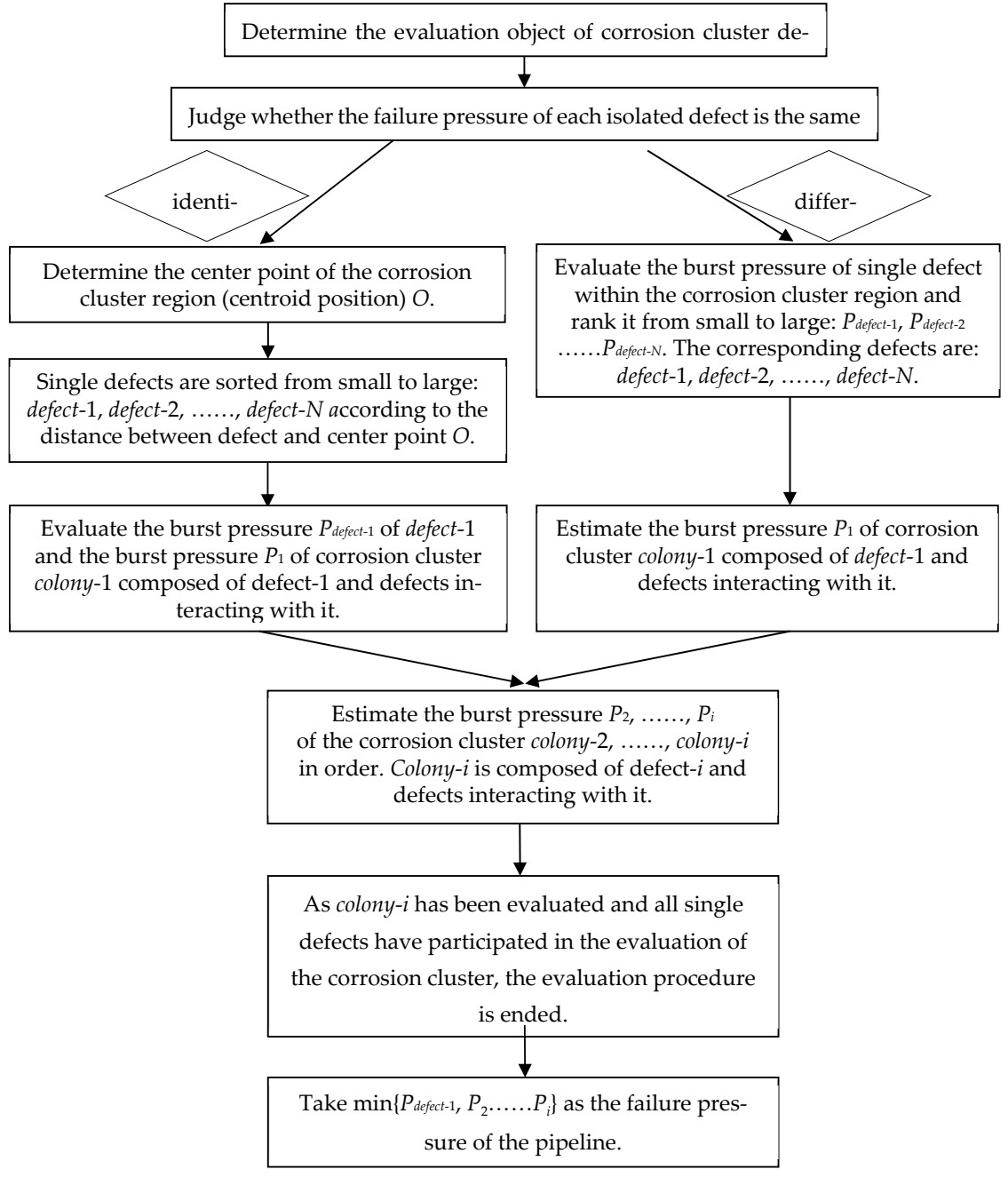

**Figure 2.** Roadmap of evaluation steps for pipeline with corrosion cluster defects.

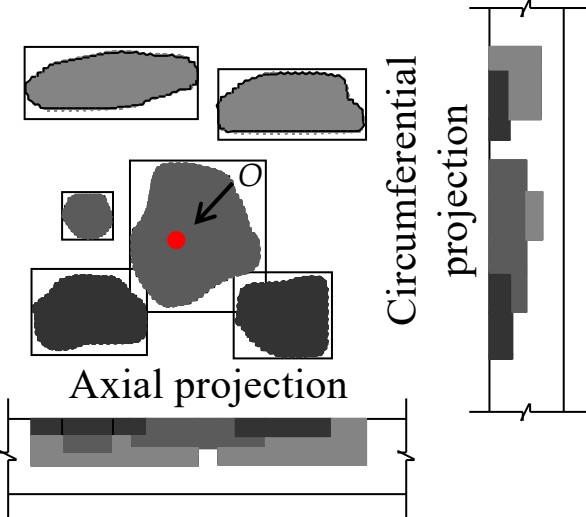

**Figure 3.** Centroid position of corrosion colony.

## 4. Verification of Assessment Method for Pipeline with Corrosion Cluster Defects

To verify the applicability and accuracy of the assessment method based on the "failure location", the DNV-RP-F101 assessment method was selected for comparison and analysis purposes.

For fear of the influence of the failure pressure calculation method on the comparison results, the two assessment methods used the same calculation model to evaluate the failure pressure of an isolated defect and colony corrosion defects.

### 4.1. Comparison of the Evaluation Cases

The assessment of corrosion cluster defects consisting of non-coincidental axial projection defects was the most challenging evaluation process. This specific type of colony corrosion was chosen for the comparative analysis between the two assessment methods.

4.1.1. Colony Corrosion Consisting of Defects with the Same Failure Pressure

Chouchaoui conducted burst tests on axially aligned defects composed of the same defects [26]. The s4lc-2 experiment was selected for scrutiny. The arrangement of corrosion cluster defects for test s4lc-2 is shown in Figure 4, and the defects are numbered as defect 1, defect 2, . . ., and defect 5.

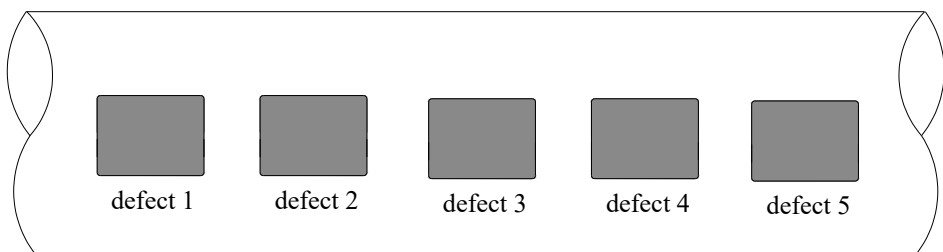

**Figure 4.** Adjacent defects for s4lc-2 [26].

For the DNV-RP-F101 assessment method, the evaluation of a single defect was required first, and then the evaluation of the interacting defects was conducted. The interacting defects were composed of two, three, four, or five individual defects. The evaluation cases composed of two defects were as follows: defect 1 and 2, defect 2 and 3, defect 3 and 4, and defect 4 and 5. The evaluation cases composed of three defects were as follows: defect 1, defect 2 and 3, defect 2, defect 3 and 4, defect 3, and defect 4 and 5. The evaluation cases composed of four defects were as follows: defect 1; defect 2; defect 3 and

4; defect 2; and defect, 3, 4, and 5. The evaluation cases composed of five defects were as follows: defect 1, defect 2, defect 3, and defect 4 and 5. The evaluation method included one single defect case and ten interacting defect cases. The number of evaluation cases to be calculated was 11.

As the size of each defect was the same and the difference was within 3%, the failure pressure could be evaluated according to the steps of the corrosion cluster composed of defects with similar internal-pressure-bearing capacities proposed in Section 3. For test s4lc-2, the failure location was identified as defect 3 based on the centroid position. This selection of failure position was confirmed with the experimental results, which also exhibited failure at the same location. Therefore, the correctness of the chosen failure location was verified. Take defect 3 as the central point and all other corrosion defects to interact with defect 3. Then, the estimating case included defect 1, 2, 3, 4, and 5. All single defects participated in this evaluation, and the assessment method was completed.

Table 3 illustrates the calculation cases of two assessment methods. The new assessment method led to a significant reduction in the number of evaluation cases that was nine less cases compared with those of the traditional method.

**Table 3.** Comparison of assessment methods for s4lc-2.

| Assessment Methods | Number of Evaluation Cases | Evaluation Cases | | | | |
|---|---|---|---|---|---|---|
| | | Single Defect | Two Interacting Defects | Three Interacting Defects | Four Interacting Defects | Five Interacting Defects |
| DNV-RP-F101 | 11 | 1 | 1–2; 2–3 3–4; 4–5 | 1–2–3; 2–3–4 3–4–5 | 1–2–3–4 2–3–4–5 | 1–2–3–4–5 |
| central failure position | 2 | 1 | - | - | - | 1–2–3–4–5 |

Note: "1–2–3" represents a corrosion cluster composed of defect 1, 2, and 3.

4.1.2. Colony Corrosion Consisting of Defects with Different Failure Pressures

Sun M and Fang H et al. [41] analyzed the strain change of colony corrosion composed with axially arranged defects under internal pressure, and relevant data were proposed. Therein, the test IDTS-X52 was adopted for verification, which is shown in Figure 5. IDTS-X52 consists of four different types of defects, with a failure pressure of 17.42 MPa, in which the difference between the failure pressure of each defect is beyond 3%. Defect 1 was a moderate and short corrosion defect. Defect 2 was a shallow and long corrosion defect. Defect 3 was a severe and short corrosion defect. Defect 4 was a moderate and extremely long corrosion defect. The spacing between defect 1 and 2 and defect 3 and 4 was 20 mm. The spacing between defect 2 and 3 was 25 mm. All adjacent defects interacted with each other. Similar to test s4lc-2, 10 evaluation cases were calculated based on DNV-RP-F101. The ten evaluation cases included four single defect cases (four single defects with different sizes) and six interacting defect cases (three cases with two defects, two cases with three defects, and one case with four defects).

Based on the assessment of four separate defects, it was determined that the failures occurred in the following order from the lowest to highest failure pressures: defect 3, 4, 1, then 2. The corrosion cluster was most likely to fail at defect 3. The actual experimental results also verified this conclusion. Firstly, the failure pressure was assessed with this failure position, i.e., defect 3 as the center point. In order to fully understand the impact of corrosion defects 2 and 4 on defect 3, it was essential to assess the overall interaction of these three defects in the 2–3–4 case. If one continued to evaluate the failure pressure centering on the second possible failure location, namely, defect 4, then interacting defect case 3–4 needed to be evaluated. Then, the failure pressure evaluation was carried out centering on the third possible failure location, namely, defect 1, and interacting defect

case 1–2 was evaluated. Then, all the corrosion defects were evaluated and the assessment process was finalized.

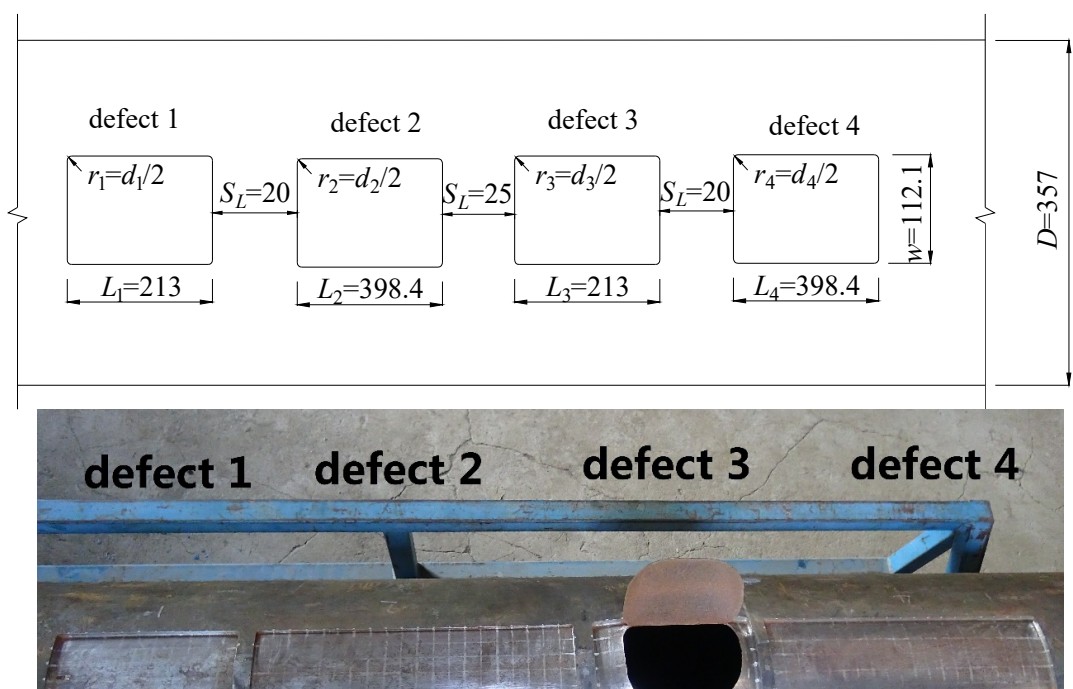

**Figure 5.** Defect number for IDTS-X52 [41].

Table 4 displays the computed instances of both evaluation methods. A comparative analysis of the two methods demonstrated that the novel assessment method reduced the number of assessment cases by three.

**Table 4.** Comparison of assessment methods for IDTS-X52.

| Assessment Methods | Number of Evaluation Cases | Evaluation Cases | | | |
|---|---|---|---|---|---|
| | | Single Defect | Two Interacting Defects | Three Interacting Defects | Four Interacting Defects |
| DNV-RP-F101 | 10 | 1; 2; 3; 4 | 1–2; 2–3 3–4 | 1–2–3; 2–3–4 | 1–2–3–4 |
| central failure position | 7 | 1; 2; 3; 4 | 1–2; 3–4 | 2–3–4 | - |

*4.2. Comparison of the Evaluation Results*

4.2.1. Burst Test of Two Contiguous Defects

The two assessment methods were analyzed based on the test of a pipeline with two contiguous defects conducted by Chouchaoui [25,26] and Benjamin [20,21].

For the experiment on the pipeline with two defects, DNV-RP-F101 and the assessment method based on "central failure position" needed to be carried out with two evaluation cases, i.e., one single-defect case and one double-interaction-defect case, respectively. The evaluation results are listed in Table 5. The results of the two assessment methods were completely consistent, and the error was same.

**Table 5.** Comparison of two corrosion defect assessment methods.

| Test | Failure Pressure $P_T$ (MPa) | DNV-RP-F101 | | | Central Failure Position | | |
|---|---|---|---|---|---|---|---|
| | | Result $P_f$ (MPa) | Number of Evaluation Cases | Error (%) | Result $P_f$ (MPa) | Number of Evaluation Cases | Error (%) |
| s1lc-2 | 17.61 | 19.14 | 2 | 8.72 | 19.14 | 2 | 8.72 |
| s2co-1 | 16.64 | 20.37 | 2 | 22.46 | 20.37 | 2 | 22.46 |
| s3co-1 | 15.95 | 19.69 | 2 | 23.46 | 19.69 | 2 | 23.46 |
| IDTS 3 | 20.314 | 19.14 | 2 | 5.76 | 19.14 | 2 | 5.76 |
| IDTS 4 | 21.138 | 21.66 | 2 | 2.47 | 21.66 | 2 | 2.47 |
| IDTS 5 | 20.873 | 18.70 | 2 | 10.42 | 18.70 | 2 | 10.42 |

Note: Error = $|P_f - P_T|/P_T \times 100\%$ where $P_f$ is the predicted failure pressure and $P_T$ is the actual experimental burst pressure.

#### 4.2.2. Test of Pipeline with Multiple Corrosion Defects
#### Verification of Axially Aligned Defects

Chouchaoui conducted burst tests on axially aligned defects, and the size of a single defect was identical [26]. The number of defects in s3lo-2, s4lo-2, s2lc-2, and s3lc-1 was three, and the number of defects in s4lc-2 was five. For the IDTS-X52 test, the dimensions of individual defects were different. The results of the two assessment methods are shown in Table 6. Compared with DNV-RP-F101, the calculation results of the assessment method based on the failure location were consistent, and calculating the cases was simpler. The errors of the two methods were positively correlated (Cov > 0), and the average values were consistent (mean = 3.74%).

**Table 6.** Comparison of assessment methods for failure pressure (axially aligned).

| Test | Failure Pressure $P_T$ (MPa) | DNV-RP-F101 | | | Central Failure Position | | |
|---|---|---|---|---|---|---|---|
| | | Result $P_f$ (MPa) | Number of Evaluation Cases | Error (%) | Result $P_f$ (MPa) | Number of Evaluation Cases | Error (%) |
| S3lo-set 2 | 14.84 | 15.67 | 6 | 5.57 | 15.67 | 2 | 5.57 |
| S4lo-set 2 | 15.53 | 15.72 | 6 | 1.20 | 15.72 | 2 | 1.20 |
| S2lc-set 2 | 15.11 | 15.88 | 6 | 5.12 | 15.88 | 2 | 5.12 |
| S3lc-set 1 | 15.67 | 16.94 | 6 | 8.12 | 16.94 | 2 | 8.12 |
| S4lc-set 2 | 15.25 | 14.89 | 15 | 2.35 | 14.89 | 2 | 2.35 |
| IDTS-X52 | 17.42 | 17.42 | 10 | 0.06 | 17.42 | 7 | 0.06 |
| mean | | - | - | 3.74 | - | - | 3.74 |
| Cov | | | | | 7.73 | | |

Note: Error = $|P_f - P_T|/P_T \times 100\%$.

#### Verification of Circumferentially Aligned Defects

Chouchaoui conducted burst tests on circumferentially aligned defects, and the size of a single defect was identical [25]. The number of defects in s3cc-1 was three, and the number in s4cc-2 was five. The results of the two assessment methods were completely consistent, and the error was also the same, as shown in Table 7.

#### Verification of Complex-Distributed Defects

Benjamin [20,21] conducted a burst test for complex-distributed defects. The experiment included pipelines with three, four, five, and nine defects.

Table 8 shows the comparison of two assessment methods for complex-distributed defects. The evaluation results and the number of evaluation cases of the two methods were the same.

**Table 7.** Comparison of assessment methods for failure pressure (circumferentially aligned).

| Test | Failure Pressure $P_T$ (MPa) | DNV-RP-F101 | | | Central Failure Position | | |
|------|------|------|------|------|------|------|------|
| | | Result $P_f$ (MPa) | Number of Evaluation Cases | Error (%) | Result $P_f$ (MPa) | Number of Evaluation Cases | Error (%) |
| S3cc-set 1 | 19.27 | 19.35 | 2 | 0.43 | 19.35 | 2 | 0.43 |
| S4cc-set 2 | 19.44 | 19.35 | 2 | 0.48 | 19.35 | 2 | 0.48 |

Note: Error = $\left| P_f - P_T \right| / P_T \times 100\%$.

**Table 8.** Comparison of assessment methods for failure pressure (complex distribution).

| Test | Failure Pressure $P_T$ (MPa) | DNV-RP-F101 | | | Central Failure Position | | |
|------|------|------|------|------|------|------|------|
| | | Result $P_f$ (MPa) | Number of Evaluation Cases | Error (%) | Result $P_f$ (MPa) | Number of Evaluation Cases | Error (%) |
| IDTS 6 | 18.656 | 16.19 | 2 | 13.20 | 16.19 | 2 | 13.20 |
| IDTS 7 | 18.772 | 16.60 | 2 | 11.56 | 16.60 | 2 | 11.56 |
| IDTS 9 | 23.06 | 20.80 | 2 | 9.80 | 20.80 | 2 | 9.80 |
| IDTS 10 | 23.23 | 20.90 | 2 | 10.02 | 20.90 | 2 | 10.02 |
| IDTS 11 | 21.26 | 18.53 | 2 | 12.83 | 18.53 | 2 | 12.83 |
| IDTS 12 | 20.16 | 16.79 | 2 | 16.73 | 16.79 | 2 | 16.73 |

Note: Error = $\left| P_f - P_T \right| / P_T \times 100\%$.

*4.3. Comprehensive Comparison of Two Assessment Methods*

The comparative analysis of the two methods was mainly carried out from three aspects: the operability of evaluation for a long-distance pipeline, evaluation accuracy, and computational complexity.

(1) The operability of evaluation for a long-distance pipeline

For the long-distance pipelines, as the internal-pressure-bearing capacity was evaluated with DNV-RP-F101, it was necessary to partition the whole corrosion region of the pipeline many times, as shown in Figure 6 ($Z = 360\sqrt{\frac{t}{D}}$ (degrees)).

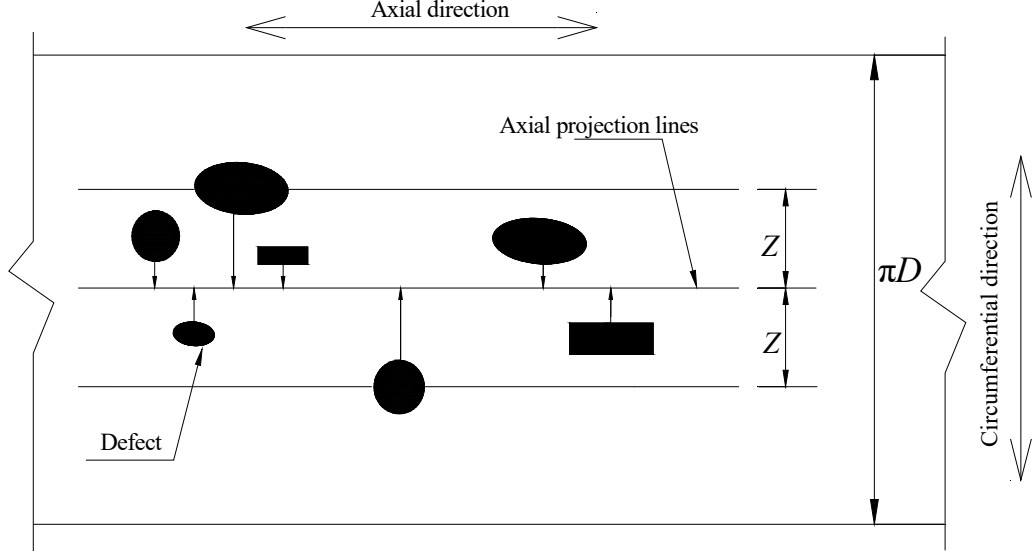

**Figure 6.** Projection of interacting defects.

The DNV-RP-F101 assessment method needed to evaluate the failure pressure of each projection line and take the minimum value as the failure pressure of the pipeline.

The accuracy of this method was affected by the axial length and projection line position, and the defects in the overlapping area were repeatedly evaluated. As a comparison, the assessment method based on failure location did not need to divide the pipeline corrosion area but only needed to sort the failure possibility of each defect and judge the existence of the interaction, which was simpler to operate.

(2)     Evaluation accuracy and computational complexity.

The comparative analysis of the actual burst test of a pipeline with corrosion cluster defects in Section 4.2 showed that the error of the new assessment method was consistent with that of the DNV-RP-F101 assessment method, and the evaluation accuracy was the same.

As the failure pressure of all individual defects is inconsistent and the axial projection does not coincide, the cases calculated by the two evaluation methods reach their maximum, and the colony corrosion defects is shown in Figure 7. The evaluation cases of DNV-RP-F101 for this defect distribution mode included $N$-times the evaluation cases of a single defect, $N − 1$ times the evaluation cases of two interacting defects, $N − 2$ times the evaluation cases of three interacting defects, . . ., and one evaluation case of $N$ interacting defects. A total of $1 + 2 + 3 + \ldots N = \frac{N(1+N)}{2}$ cases needed to be evaluated.

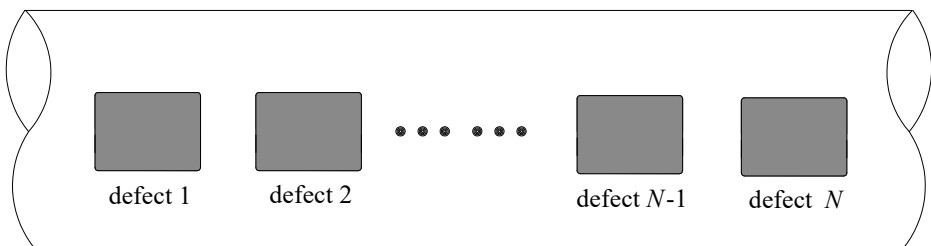

**Figure 7.** Corrosion cluster defects with non-coincidental axial projections.

For this corrosion defect distribution mode, the first condition was that the corrosion defects affected each other, as shown in Section 4.1.1. For this condition, $N + 1$ cases needed to be calculated for the assessment method based on the failure location: $N$-times evaluation cases of a single defect and one evaluation case of $N$ interacting defects. The second condition was that each single defect only affected the adjacent defects. At the same time, the two defects with the largest burst pressures and the two defects with the smallest internal-pressure-bearing capacities were located at both ends of the corrosion cluster defects, such as the example in Section 4.1.2. In this case, the number of evaluation cases calculated with the assessment method based on the failure position reached the maximum value. The cases that needed to be calculated are as follows: $N$-times evaluation cases of a single defect and $N − 1$ times evaluation cases of two or three interacting defects. In total, $2N − 1$ cases were required to be calculated.

Table 9 compares the evaluation cases of the two methods for different corrosion clusters composed of $N$ defects. It can be seen from the table that the assessment method based on failure location only needed one calculation of interacting defects in the evaluation colony of corrosion in which every two defects all interact, which was simpler and more efficient. In evaluating corrosion cluster defects with a non-overlapping axial projection, the number of evaluation cases was much less than that of the DNV-RP-F101 assessment method. Only as the failure pressure was evaluated for corrosion clusters composed of the same defects whose axial projections coincided with each other were the number of evaluation cases based on the failure location the same as that of DNV-RP-F101.

**Table 9.** Number of calculated working conditions for group corrosion.

| Arrangement Mode | Characteristics of Single Defect | Interaction Mode between Corrosion Defects | Number of Evaluation Cases | |
|---|---|---|---|---|
| | | | DNV-RP-F101 | Central Failure Position |
| Axial projection did not overlap | The failure pressure was different | Every two all interact | $\frac{N(1+N)}{2.5}$ | $N+1$ |
| | | Every two do not all interact | | $\leq 2N-1$ |
| | The failure pressure was the same | Every two all interact | $\frac{N(N-1)}{2}+1$ | $2$ |
| | | Every two do not all interact | | $\leq N$ |
| All axial projection coincidence | The failure pressure was different | - | $N+1$ | $N+1$ |
| | The failure pressure was the same | - | $2$ | $2$ |

## 5. Conclusions

The failure modes of interacting defects of the same or different sizes were analyzed, and the failure positions of corrosion cluster defects with different modes were obtained. Combined with the failure location of a corroded pipeline under internal pressure, an improved failure pressure assessment method for the colony of corrosion defects based on failure location is put forward in this paper. The conclusions that can be drawn are as follows:

(1) For the new assessment method proposed in this paper, the defects were sorted according to the probability of being in the failure position from the largest first to the smallest, and then the failure pressure was evaluated with the arranged defect as the "central failure position", in order. As all individual defects were involved in the evaluation, the failure pressure assessment procedure finished. This assessment method evaluated the interacting defects most likely to fail based on failure location, and this method did not omit the evaluation case with the minimum failure pressure.

(2) The evaluation error of the assessment method based on failure location did not change compared with DNV-RP-F101, but its evaluation cases were reduced. Its operability was concise and had strong applicability. Its accuracy and smaller number of evaluation cases made the new assessment method more applicable and far more effective. The accurate evaluation method of failure pressure for individual defects and interaction judgment criteria was the prerequisite for the implementation of the new method.

**Author Contributions:** Conceptualization, H.Z.; Methodology, M.S.; Software, B.L.; Validation, K.Z.; Data curation, J.Z.; Writing—original draft, Y.Z. All authors have read and agreed to the published version of the manuscript.

**Funding:** This project was supported by the Key Scientific Research Projects of Colleges and Universities in Henan Province (23A560013), the National Key R&D Program of China (No. 2022YFC3801000), the Henan Provincial Youth Science Foundation (232300421328), and the Central Plains Technology Innovation Leading Talent Project (Central Plains Talent Program: 234200510014).

**Data Availability Statement:** All data generated or analyzed during this study are included in the manuscript file and the published paper. And the datasets used or analyzed during the current study are available from the corresponding author upon reasonable request.

**Conflicts of Interest:** The authors declare no conflict of interest.

## Nomenclature

| | |
|---|---|
| $colony - i$ | $colony - i$ formed by $defect - i$ and defects interacting with it |
| $defect - i$ | defect with failure pressure ranking $i$ |
| $D$ | outside diameter of the pipe |
| $N$ | number of individual defects |
| $P_i$ | failure pressure of corrosion cluster $colony - i$ |
| $P_f$ | failure pressure of the corroded pipeline |
| $P_{\text{defect}-1}$ | failure pressure of $defect - 1$ |
| $P_T$ | the actual experimental burst pressure |
| $r_i, d_i, L_i$ | chamfer radius, depth, and length of defect $i$ |
| $S_L$ | axial spacing between adjacent defects |
| $t$ | wall thickness of the pipe |
| $Z$ | circumferential spacing of projection lines |

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
