# Peer review of "Study on Assessment Method of Failure Pressure for Pipelines with Colony Corrosion Defects Based on Failure Location"

_processes, doi:10.3390/pr11113134_

Round 1

Reviewer 1 Report

Comments and Suggestions for Authors

The paper experimentally studied the failure modes of interacting defects with the same or different size. The paper is good. The revision:

1) Please add a notation list.

2) Fig. 5 and Table 3 need more discussion in the text.

3) In order to provide a more comprehensive literature review, the authors should cite and discuss the following relevant papers in their revised manuscript. The papers investigated the failure modes and capacity of steel tubular members.

-Static capacity of collar plate reinforced tubular X-connections subjected to compressive loading: study of geometrical effects and parametric formulation. Ships and Offshore Structures2021;16(1), pp.54-69.

-Static capacity of tubular X-joints reinforced with collar plate subjected to brace compression. Thin-Walled Structures2017;119, pp.256-265.

4)  in Table 6, please add the mean and CoV.

5) line 133-150 should be rewritten. The equations are not properly written.

6) There should be referenced in the equations, figures, and tables if they are taken from somewhere else. Also, All formulas should be checked, and Italic forms in formulas should be followed in the text.
7) Please give some suggestion to reduce the error of assessment method based on failure location, compared with DNV-RP-F101.

Reviewer 2 Report

Comments and Suggestions for Authors

The reviewed work has great practical potential. The occurrence of failures related to the presence of corrosion mechanisms may lead to large losses and environmental hazards. The proposed analysis method is innovative and should be used in practice. The only thing missing in the work is the reference of the analysis used to the API 579-1/ASME FFS-1 - Fitness-For-Service (FFS) standard.

Reviewer 3 Report

Comments and Suggestions for Authors

1. The following reference are recommended for the authors to grasp the current research status:  https://www.sciencedirect.com/science/article/pii/S2352854023000517.

2. Minor revisions needs to be conducted regarding to the English language of the manuscript.

3. How to compare the performance of the proposed method with the existing DNV-RP-F101? 

Comments on the Quality of English Language

Minor revisions are needed for improving the quality of English language of this manuscript. The manuscript as a whole is readable.

Round 2

Reviewer 1 Report

Comments and Suggestions for Authors

Ready to publish.